# Resistance to Fusarium Head Blight, Kernel Damage, and Concentration of *Fusarium* Mycotoxins in Grain of Winter Triticale (x *Triticosecale* Wittmack) Lines

**Tomasz Góral** [1,*] , **Halina Wiśniewska** [2,*], **Piotr Ochodzki** [1], **Adriana Twardawska** [2] and **Dorota Walentyn-Góral** [1]

[1] Department of Plant Pathology, Plant Breeding and Acclimatization Institute—National Research Institute, Radzików, 05-870 Błonie, Poland; p.ochodzki@ihar.edu.pl (P.O.); d.walentyn-goral@ihar.edu.pl (D.W.-G.)
[2] Institute of Plant Genetics, Polish Academy of Sciences, 34 Strzeszyńska Str., 60-479 Poznań, Poland; atwa@igr.poznan.pl
* Correspondence: t.goral@ihar.edu.pl (T.G.); hwis@igr.poznan.pl (H.W.); Tel.: +48-22-733-4636 (T.G.)

**Abstract:** Fusarium head blight (FHB) can cause contamination of cereal grain with mycotoxins. Triticale is also infected with FHB; however, it is more resistant than wheat to head infection. The aim of this study was to identify triticale lines that combine low head infection with low toxin contamination. Resistance to FHB of 15 winter triticale and three winter wheat lines was evaluated over a three-year experiment established in two locations. At the anthesis stage, heads were inoculated with *Fusarium culmorum* isolates. The FHB index was scored and the percentage of *Fusarium*-damaged kernels (FDKs) assessed. The grain was analysed for type B trichothecenes (deoxynivalenol and derivatives, nivalenol) and zearalenone content. The average FHB index was 10.7%. The proportion of FDK was 18.1% (weight) and 21.6% (number). An average content of deoxynivalenol amounted to 7.258 mg/kg and nivalenol to 5.267 mg/kg. In total, it was 12.788 mg/kg of type B trichothecenes. The zearalenone content in the grain was 0.805 mg/kg. Relationships between FHB index, FDK, and mycotoxin contents were statistically significant for triticale lines; however, they were stronger for FDK versus mycotoxins. Triticale lines combing all types of FHB resistance were found, however the most resistant ones were less resistant that wheat lines with the *Fhb1* gene.

**Keywords:** deoxynivalenol; *Fusarium culmorum*; Fusarium head blight; nivalenol; triticale; trichothecenes; zearalenone



## 1. Introduction

Triticale (x *Triticosecale* Wittmack) is the first synthetic successful amphiploid cereal, which originated in 1874, from hybridization of hexaploid wheat (*Triticum aestivum* L.) and rye (*Secale cereale* L.) [1]. The intergeneric synthetic hybrids combined the complementary traits of both parental species—the high yielding capacity of wheat and the stress tolerance of rye. Because triticale compromises the beneficial agronomic traits of wheat and the resistance to environmental stresses of rye, at the end of 20th century, the production of this cereal had significantly grown [2,3]. Triticale grain production increased twofold, from six M tons in 1995 to almost 13 M tons in 2018 worldwide. Similarly, in Poland, production increased from two M tons to four M tons over 23 years. Although the arable land for wheat (2,417,227 ha) is twice as large as for the triticale species (1,287,969 ha), Poland leads in the production of this crop worldwide. In Poland, triticale is cultivated twice as much as in Germany and four times more than in Belarus, France, Spain, and China [4]. Nowadays, triticale plants are used in a variety of ways, mostly as grain intended for feed production [3]. Additionally, during spring, the land cultivated for triticale is used for pasture, as fresh feed for livestock or for hay and silage. Recently triticale has also been cultivated for biofuels [3] and bioethanol [5] as well as for food production such as bread or cakes [6]. As farm animals consume triticale grain and products for human consumption are also

made from triticale grains, it is important to maintain good quality grains, especially in the case of detrimental compounds content e.g., mycotoxins.

Recently, a decrease in triticale resistance to pathogens of the *Fusarium* genus has been observed. Fusarium head blight (FHB) is a destructive disease of wheat and triticale, which causes significant loss of yield and quality as well as the accumulation of hazardous mycotoxins in the grain. Numerous species of *Fusarium* have been associated FHB in triticale and wheat, especially *Fusarium culmorum* (W.G. Smith) Sacc. and *F. graminearum* Schwabe [7,8]. Suitable, rainy weather during the flowering and soft dough stages of kernel development plays a crucial role for the establishment and severity of the disease [9–13]. *Fusarium* develops in the infected flower, then overgrows to the next ones and afterwards spreads through the rachis along the whole head. *Fusarium* colonise chaff and kernels in the ear, damaging them at different levels. It reduces grain yield and grain quality by contaminating grain with mycotoxins [14,15]. FHB resistance consists of several mechanisms (types): resistance to initial infection (type 1), resistance to *Fusarium* spread within the spike (type 2), resistance to kernel infection (type 3), tolerance to accumulated toxins (type 4), resistance to accumulation of *Fusarium* toxins in the grain (chemical modification/synthesis inhibition = type 5) [16–18].

Resistance to FHB is a quantitative feature [19]. The presence of several quantitative trait loci (QTL) associated with FHB resistance has been reported. Loci associated with FHB resistance originate from various types of Asian spring wheats, e.g., Sumai 3, Wuhan 1, Nyubai, Wangshuibai, and Nobeokbozukomugi [20].

Breeding for improved FHB resistance is laborious task as this trait is quantitative in nature. It is greatly affected by the genetic characteristics of the host plant and fungal pathogens. Environmental conditions, mostly temperature and rainfall, from anthesis to the soft dough stage also have a substantial influence on FHB's development and make efficient selection a difficult task [9–11,14,16,21]. Breeding of cultivars resistant to FHB play a key role in disease control and the prevention of mycotoxin contamination [22–24].

*Fusarium* species produce secondary metabolites (mycotoxins) belonging to different chemical groups. Those most often found in cereal grains are type B trichothecenes: deoxynivalenol (DON) and nivalenol (NIV); type A trichothecenes: T-2 and HT-2 toxins; zearalenone (ZEN) and moniliformin [8]. They are extremely stable, non-metabolizable compounds with great harm to humans and animals [25]. Grain contamination with mycotoxins is found even when no reduction in yield is observed [26]. Recent research shows that triticale, like wheat, is significantly threatened by FHB and the critical accumulation of mycotoxins in grain [27,28].

A large issue with modern cereal cultivation is the presence of *Fusarium* mycotoxins, such as NIV, DON, and ZEN in grain [29]. In 2007, the European Commission set the maximum level of deoxynivalenol in common wheat, triticale and rye grains at 1.250 mg/kg and zearalenone at 0.100 mg/kg. The maximum level of DON was also established in flour at 0.750 mg/kg and in bread at 0.500 mg/kg level. The maximum ZEN content in flour was set at 0.075 mg/kg and in bread at 0.050 mg/kg (Commission Regulation (EC) no. 1126/2007) [30]. Whereas, in feed production, the lowest level of DON is required in pig feed production at 0.9 mg/kg and ZEN in piglet and sow feed production at 0.1 mg/kg (Commission Recommendation (EC) no. 2006/576) [31].

Small grain cereals differ considerably in their resistance to FHB and accumulation of mycotoxins. Research, which compared triticale and its parental forms in terms of FHB and deoxynivalenol content, revealed that triticale and rye had the lowest FHB severity and kernel damage, whereas the lowest deoxynivalenol concentration was obtained in rye followed by triticale and wheat species [29]. However, in an earlier study Langevin et al. [32] found that triticale responded similarly to wheat to the point inoculation with *F. graminearum*. In our study, we observed that triticale lines were more resistant to FHB and kernel damage than wheat but were similar in regard to *Fusarium* toxins accumulation [27]. Depending on environmental conditions and genotype, triticale could accumulate high amounts of trichothecene toxins [15,33].

The aim of this study was to compare the susceptibility of winter triticale lines to Fusarium head blight (head infection and kernel damaged) and accumulation of mycotoxins (type B trichothecenes and zearalenone) in grain. Additionally ergosterol concentration in grain was analysed. Ergosterol is main sterol compound of fungal cell membranes and can be a measure of mycelium amount in cereal kernels [34].

Experiments were established under different conditions in two locations in Poland. Plants were inoculated with *Fusarium culmorum* isolates. We studied different types and mechanisms of resistance: resistance to head infection, resistance to kernel damage, tolerance to accumulated toxins, and resistance to accumulation of *Fusarium* toxins in the grain.

## 2. Materials and Methods

### 2.1. Plant Materials

Plant material comprised 18 triticale and wheat lines of winter type (Table 1).

**Table 1.** Triticale and wheat lines used for the research.

| Line | Type | Pedigree |
|---|---|---|
| Meloman | triticale cultivar | MAH 24050-12 × Todan |
| BOH 534-4 | triticale breeding line | BOH 9-3/4 × Clever (wheat) |
| BOH 537-2 | triticale breeding line | BOH 9-3/4 × LAD 965/98 |
| BOH 898-1 | triticale breeding line | MAH 26699-21 × Moderato |
| BOHD 1025-2 | triticale breeding line | BOH 303 × LAD 180/00 |
| BOHD 1062-2 | triticale breeding line | DH 265 × (Salvo × Prado) |
| DANKO 6 (2014) | triticale breeding line | DED 145/05 × Tulus |
| DANKO 9 (2013) | triticale breeding line | DED 357/00 × ADM 8 |
| DL 446/08 | triticale breeding line | (LAD 950/99 × Moderato) × Grenado |
| DL 593/07 | triticale breeding line | (DED 355/97 × Grenado) × Woltario |
| DS.1238 | triticale breeding line | CHD 651/96 × DED 396/95 |
| DS.9 | triticale breeding line | Lasko × Woltario |
| LD 121/08 | triticale breeding line | DT 1282/00 × Kitaro |
| MAH 33544-4 | triticale breeding line | MAH 23175-1/20/24 × MAH 25547-2 |
| MAH 33881-1/3 [a] | triticale breeding line | MAH 24050-12 × Todan |
| DL 325/11/3 | wheat breeding line | (Figura x Koch) × CHD 752/01 |
| S10 | wheat FHB resistant line | Korweta × Sumai 3 |
| S32 | wheat FHB resistant line | Turnia × Sumai 3 |

[a] registered as Trefl cultivar.

Polish breeding companies (DANKO Hodowla Roślin, Choryń, Poland; Hodowla Roślin Strzelce, Choryń, Poland) developed the above triticale lines. They were selected based on low *Fusarium* head infection investigated in two environments [15]. Wheat breeding line was susceptible to FHB [35]. Two FHB resistant lines were developed in Plant Breeding and Acclimatization Institute NRI and carried *Fhb1* resistance gene [36].

### 2.2. Fungal Material for Inoculation

The material for inoculum production consisted of three isolates of *Fusarium culmorum* (W.G.Sacc.). KF 846 (DON chemotype) and KF 350 (NIV chemotype) originated from the collection of the Institute of Plant Genetics Polish Academy of Sciences (Poznań, Poland). The ZFR 112 (DON chemotype, producing high amounts of ZEN in vitro) originated from the collection of Plant Breeding and Acclimatization Institute, National Research Institute (NRI) (Radzików, Poland) [37].

Isolates were incubated on autoclaved wheat grain in glass Erlenmeyer flasks (300 mL) for one week at 20 °C in darkness and then exposed to near UV light (360 nm) under a 16 h photoperiod for 3 weeks at 15 °C. Flask were manually shaken daily to avoid the grain sticking together and to break-up the mycelial clumps. The mycelium-colonised grain with visible spore masses was air dried and stored in a refrigerator at 2–5 °C. Prior to the inoculation, the grain with *F. culmorum* spores was soaked in distilled water for approximately

2 h. Next, the suspension was filtered through a double cheesecloth layer to harvest spores and remove grains and mycelium. The conidial suspensions from three *F. culmorum* isolates were adjusted to 500,000 spores/mL using a haemocytometer (BRAND GmbH + Co. KG., Wertheim, Germany). Equal volumes of suspension from the three isolates were mixed.

### 2.3. Field Experiment

A three-year field experiment (2016, 2017, and 2018) was established in two locations. The first experimental field location was the Institute of Plant Genetics Polish Academy of Sciences in Cerekwica (30 km north-west from Poznań, Poland; 82 m above sea level; GPS coordinates 52°31′21.3″ N, 16°41′19.1″ E). The second experimental field location was the Plant Breeding and Acclimatization Institute, NRI in Radzików (central Poland; 87 m above sea level; GPS coordinates 52°12′45.4″ N, 20°37′59.2″ E). Experiments were established as randomized block designs. Triticale lines were sown in 1 m$^2$ (Radzików) or 0.5 m$^2$ (Cerekwica) plots in four replicates/blocks. Sowing dates were within the range from the last week of September to the first week of October. In both locations conventional tillage was applied.

In Radzików in all three years pre-crop was oilseed rape. Artificial fertilizers were applied according to standard agricultural practices in IHAR-PIB. In the autumn 3 dt/ha of Polifoska 6 (NKP(S) 6-2-30-(7)) fertilizer (Grupa Azoty Zakłady Chemiczne "Police" S.A., Police, Poland) was applied (N—18 kg/ha, P—60 kg/ha, K—90 kg/ha). In the spring, after the start of vegetation ammonium nitrate fertilizer (Grupa Azoty Zakłady Azotowe "Puławy" S.A., Puławy, Poland) was applied in an amount providing 70 kg N/ha. Weeds and pests were controlled with herbicides and insecticides. After sowing weeds were controlled with herbicide Maraton 375SC (BASF SE, Ludwigshafen, Germany) (iso-proturon + pendimethalin) in a dose of 4 L/ha. In spring, weeds and rape self-seeders were controlled using herbicide Attribut 70SG (Bayer CropScience AG, Monheim, Germany) (propoxycarbazone-sodium) in a dose of 60 mg/ha. Cereal leaf beetle and aphids were controlled with Fastac Active 050ME (BASF SE, Ludwigshafen, Germany) (alpha-cypermethrin) in a dose of 250 mL/ha. No fungicides were applied.

In Cerekwica pre-crop was oilseed rape (2017, 2018) or lacy phacelia (2016). In the autumn 4 dt/ha of Polifoska 5 (NPK(MgS) 5-15-30-(2-7)) fertilizer (Grupa Azoty Zakłady Chemiczne "Police" S.A., Police, Poland) was applied (N—20 kg/ha, P—60 kg/ha, K—120 kg/ha). In the spring, after the start of vegetation ammonium nitrate fertilizer (Grupa Azoty Zakłady Azotowe „Puławy" S.A., Puławy, Poland) was applied in an amount providing 70 kg N/ha. After sowing weeds were controlled with herbicide Legato 500SC (ADAMA Polska Sp. z o.o., Warszawa, Poland) (diflufenican) in a dose of 1.5 L/ha. No fungicides were applied.

### 2.4. Inoculation Procedure

At full anthesis (65 BBCH scale [38]), triticale and wheat lines were inoculated by spraying heads with a spore suspension [39]. Full anthesis of triticale lines in particular years was as follows 2016: Cerekwica 30–31 May, Radzików 30 May–2 June; 2017; Cerekwica 1–7 June, Radzików 8–13 June; 2018; Cerekwica 22–28 May, Radzików 29 May–3 June. For three wheat lines, this stage was on average 2–3 later than the latest triticale line. Three blocks of plots were inoculated and a fourth served as a control. Inoculation was repeated three days later.

For two days after inoculation, in Cerekwica mist irrigation was applied to maintain high moisture levels on inoculated plots [37,40]. In Radzików experimental field is located about 500 m from the river valley and in 2016 and 2017 average air humidity during anthesis was at a range 60–80%. Only in 2018 (dry year in Poland) it was about at a range 50–60%.

Three weeks after inoculation, disease progress was visually evaluated as the Fusarium head blight index (FHBi):

$$\text{FHBi} = \frac{\% \text{ of head infection} \times \% \text{ of heads infected per plot}}{100} \tag{1}$$

At harvest, 20 randomly selected heads from each plot (one control and three inoculated plots) in each location were collected. Heads were threshed with a laboratory thresher at low wind speed to prevent loss of low-weight infected kernels. Additionally after threshing the individual sample, chaff was inspected for the presence of shrivelled, low-weight FHB damaged kernels.

The percentage of *Fusarium*-damaged kernels (FDK) was scored visually according to the methods described earlier [41,42]. The FDK weight in relation to the weight of the whole sample was marked as FDKw, and the FDK number in relation to the total sample size was marked as FDK#.

Reductions in the yield components caused by FHB in relation to the non-inoculated control were calculated. The components were as follow: grain yield per head, kernel number in head, and thousand kernels weight (TKW).

### 2.5. Toxins and Ergosterol Analysis

Whole grain samples were fine ground. The concentration of *Fusarium* toxins in triticale grain was analysed using the technique of gas chromatography. The type B trichothecenes (DON, 3-acetyldeoxynivalenol (3AcDON), 15-acetyldeoxynivalenol (15AcDON) and NIV) were detected. The methodology used for the extraction and detection of the samples with use of gas chromatography was described in detail by Góral et al. [27,43].

The content of ZEN was determined using a quantitative direct, competitive enzyme-linked immunosorbent assay (ELISA) AgraQuant® Zearalenone 25-1000 (LOD 20 ppb, LOQ 25 ppb) (Romer Labs GmbH, Tulln, Austria). The detailed methodology used for the quantitative analysis of ZEN was described by Góral et al. [27].

Ergosterol (ERG) was chromatographically analysed via high-performance liquid chromatography (HPLC) on a silica column using methanol. A detailed evaluation of the method is given in a paper by Perkowski et al. [34]. Samples containing 100 mg of ground grains were placed into 17 mL culture tubes, suspended in 2 mL of methanol, treated with 0.5 mL of 2 M aqueous sodium hydroxide and tightly sealed. The culture tubes were then placed within 250 mL plastic bottles, tightly sealed and placed inside a microwave oven operating at 2450 MHz and 900 W maximum output. Samples were irradiated (370 W) for 20 s, and following approximately 5 min, for an additional 20 s. After 15 min, the contents of the culture tubes were neutralized with 1 M aqueous hydrochloric acid, 2 mL MeOH were added and extraction with pentane ($3 \times 4$ mL) was carried out within the culture tubes. The combined pentane extracts were evaporated to dryness in a nitrogen stream. Before analysis samples were dissolved in 1 mL of MeOH, they were filtered through 13 mm syringe filters with a 0.45 μm pore diameter (Fluoropore Membrane Filters, Millipore, Corcaigh, Ireland) and 50 μL were injected on HPLC column. Separation was performed on a reversed phase column Nova Pak C-18 (Waters, Milford, MA, USA), $150 \times 3.9$ mm, particle size 4 μm, and eluted with methanol/acetonitrile (90:10) at a flow rate of 0.6 mL/min. Ergosterol was detected with a Waters 486 Tunable Absorbance Detector (Milford, MA, USA) set at 282 nm. The presence of ergosterol (ERG) was confirmed by a comparison of retention times and by co-injection of every tenth sample with an ergosterol standard.

### 2.6. Screening of Type 1 and Type 2 FHB Resistances

Winter triticale and wheat lines were sown in two field experiments in 2015, 2016, 2018, and 2019 in the experimental field in Radzików. Lines were sown in 1-row plots of 1 m length without replications. Spacing between rows was 30 cm. Plots were covered with polyethylene tents equipped with mist irrigation system. Mist irrigation was applied after inoculations to keep high air humidity.

In the first experiment, type 1 resistance was evaluated. At the full flowering phase (BBCH 65), the heads were sprayed with a suspension of spores of *F. culmorum* isolate KF 846 with a concentration of $10^5$ spores/mL. The number of infection points was assessed 7 days after inoculation. Spikelets showing necrotic spots were scored as one infection point. The methodology for assessing this resistance type was similar to that described by Kubo et al. [44] and the Patton-Ozkurt et al. methodology available on the US Wheat and Barley Scab Initiative web page [45].

In the second experiment, resistance of type 2 was evaluated. The methodology for assessing this resistance was similar to that described by Rudd et al. [46] and by G.H. Bai, available on the US Wheat and Barley Scab Initiative web page [47]. Heads were inoculated at the full flowering phase (BBCH 65) by placing a drop (approx. 50 μL) of a suspension of *F. culmorum* spores in the flower of the middle spikelet of labelled heads with a self-filling syringe Dosys™ classic 163/173 (Socorex ISBA SA, Ecublens, Switzerland). The suspension concentration was $50 \times 10^3$ spores/mL, which gave about 2500 spores per suspension drop. The same isolate KF 846 was used. Ten heads per line were inoculated. The severity of FHB was assessed by determining the number of infected spikelet per head 21 days after inoculation. The infection symptoms include necrosis, whitening of the spikelets, in some cases *Fusarium* sporulation in the form of a pink-orange sporodochia.

### 2.7. Statistical Analysis

Statistical analysis was made using Microsoft® Excel 2016/XLSTAT© (Version 2020.4.1.1027, Addinsoft, Paris, France) statistical software.

FHB and FDK ratings, reduction of yield components and concentration of ERG and toxins data were analysed by means of analysis of variance (ANOVA) using the XLSTAT procedure: Modelling data–ANOVA. Year effect was considered random and location and line were considered fixed. Normality of data distribution was tested with the Shapiro–Wilk test (XLSTAT procedure: Normality test). All variables were not normally distributed; hence, they were transformed with the square root (FHBi, FDKw, FDK#, yield reduction, kernel# reduction, TKW reduction) or log10 (ERG, DON, 3ADON, 15AcDON, NIV, trichothecenes (TCT B), ZEN) transformations.

The relationships between resistance type 1 and type 2, FHBi, FDK, reduction in the yield components and the ERG and mycotoxins concentrations were analysed using Pearson's correlation tests (XLSTAT procedure: Correlation tests). Prior to analysis, variables (means for 18 lines) that were not normally distributed were square root (FHBi, FDKw, FDK#, yield reduction, kernel# reduction, TKW reduction) or log10 transformed (type 1, type 2, type 1 + 2, ERG).

The data on FHB resistance (FHBi, FDKw, yield reduction, TKW reduction, ERG, DON, NIV, ZEN) measured with different units were analysed together using multivariate statistical analysis. Principal component analysis (XLSTAT procedure: Principal Component Analysis PCA) was used to show how triticale (and wheat) lines are distributed with respect to the variation described by the first two principal components and how FHB resistance variables influenced the two components. PCA results also show relations among variables measured by the angle among variable vectors.

## 3. Results

No FHB symptoms were observed on control plots. The average severity of FHB on inoculated plots was FHBi = 10.7%. It was similar in both locations and amounted to 11.3% in Radzików and 9.7% in Cerekwica (Figure 1). The range of reactions was from 0% to 64.0% in Radzików and from 2.7% to 40.0% in Cerekwica. The proportion of *Fusarium*-damaged kernels was higher in Cerekwica (FDKw = 34.9%; FDK# = 27.9%) than in Radzików (FDKw = 12.7%; FDK# = 11.4%). The range of reaction was from 0.2% to 47.3% in Radzików and from 0.3% to 84.5% in Cerekwica for FDKw and from 0.6% to 51.4% in Radzików, and from 0.2% to 88.6% in Cerekwica for FDK#.

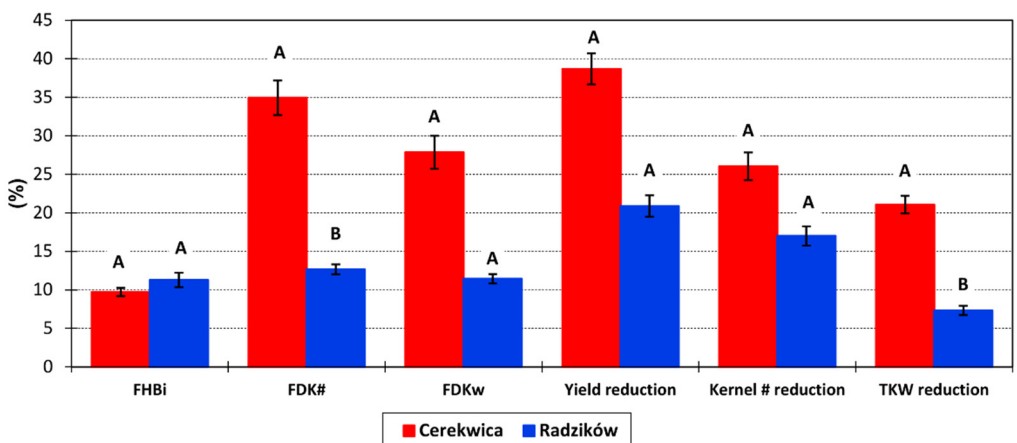

**Figure 1.** Average Fusarium head blight index (FHBi), *Fusarium*-damaged kernels proportion (FDK#—number, FDKw—weight) and reductions in the yield components (grain yield per head, kernel number in head, and 1000 kernels weight (TKW)) in two experimental locations. The data are means $\pm$ SEM ($n$ = 162). Means marked with the same letter are not significantly different at $p < 0.05$ according to Fisher's LSD test performed on transformed variables.

Over three experimental years, the FHB index amounted to 14.4% in 2016, 13.6% in 2017, and 4.0% in 2018. As regards the FDK proportion, it was 21.2% and 26.2% in 2016, 22.8% and 25.2% in 2017, and 10.0% and 13.4% in 2018 for FDKw and FDK#, respectively.

Reductions in the three studied yield components were, on average, grain yield 28.3%, kernel number 20.8%, and TKW 13.0%. In the experiment in the Cerekwica, the reductions were higher than in Radzików (Figure 1). Over the three experimental years, the reductions in yield, kernel number and TKW were as follows: 26.3%, 16.2%, and 14.9% in 2016; 27.8%, 21.8%, and 13.0% in 2017; and 30.7%, 24.3%, and 11.0% in 2018.

Concentration of ERG in grain was, on average, 8.3 mg/kg. It was similar for both locations (Figure 2). The concentration range was 1.5–38.7 mg/kg in Cerekwica and 0.6–35.2 mg/kg in Radzików. In 2016, the average ERG content in grain was 5.6 mg/kg, in 2017 8.8 mg/kg and in 2018 11.0 mg/kg.

The amount of DON in grain was, on average, 7.258 mg/kg at a range 0.028–50.330 mg/kg and the amount of NIV in grain was 5.267 mg/kg in the range 0–44.628 mg/kg. In Radzików, the concentration of DON was twice as high as in Cerekwica (Figure 2). On the contrary, the concentration of NIV in Radzików was low (0.855 mg/kg) and 10 times lower than in Cerekwica (9.679 mg/kg). Over the three experimental years, the amounts of DON and NIV were as follows: 2016—5.991 and 4.402 mg/kg; 2017—13.248 and 7.974 mg/kg; and 2018—2.536 and 3.426 mg/kg.

Acetylated derivatives of DON (e.g., 3AcDON and 15AcDON) were detected in low amounts. On average, it was 0.228 mg/kg of 3AcDON (0–1.352 mg/kg) and 0.035 mg/kg of 15AcDON (0–0.445 mg/kg). In Cerekwica, the concentration of 3AcDON and 15AcDON was 0.284 and 0.001 mg/kg, respectively, and in Radzików 0.172 and 0.068 mg/kg. Over the three experimental years, the amounts of 3AcDON and 15AcDON were as follows: 2016—0.236 and 0.019 mg/kg; 2017—0.444 and 0.078 mg/kg; and 2018—0.004 and 0.007 mg/kg.

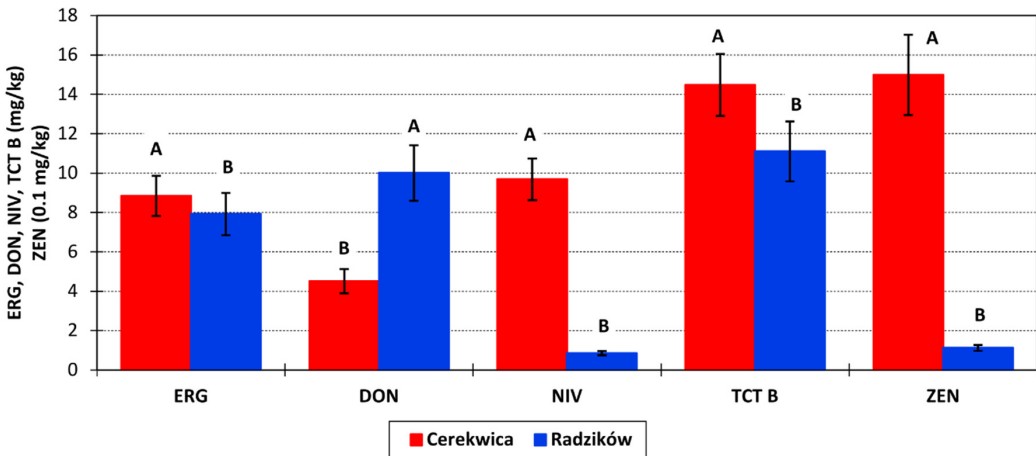

**Figure 2.** Average concentration of ergosterol (ERG), deoxynivalenol (DON), nivalenol (NIV), type B trichothecenes (TCT B), and zearalenone (ZEN) in grain of 15 triticale and 3 wheat lines in two experimental locations. Neither 3-acetyldeoxynivalenol (3AcDON) nor 15-acetyldeoxynivalenol (15AcDON) are shown. The data are means ± SEM (*n* = 54). Means marked with the same letter are not significantly different at *p* < 0.05 according to Fisher's LSD test performed on transformed variables.

The total amount of analyzed type B trichothecenes was 12.788 mg/kg at a range of 0.100–65.565 mg/kg. In Cerekwica, the amount of TCT B was 14.476 mg/kg at a range 1.453–65.565 mg/kg, and in Radzików 11.101 mg/kg at a range 0.100–53.595 mg/kg. Over three years, the average amount of TCT B was as follows: 2016—10.648 mg/kg, 2017—21.744 mg/kg, and 2018—5.973 mg/kg.

Zearalenone was detected in grain at an average amount of 0.805 mg/kg. The amount ranged from 0 to 5.055 mg/kg. ZEN was present mainly in samples from Cerekwica at amount of 1.500 mg/kg. In samples from Radzików, the concentration was 10 times lower and amounted to 0.112 mg/kg. The highest concentration of ZEN was detected in 2017 at 1.446 mg/kg, followed by 2016 at 0.900 mg/kg. In 2018, it was very low at 0.071 mg/kg.

Winter wheat lines showed the highest (DL 325/11/3) and the lowest (lines carrying *Fhb1* resistance gene) values of FHB index (Table 2). Among triticale lines, the lowest FHBi was observed for six lines: DANKO 9 (2013), LD 121/08, BOHD 1025-2, BOH 534-4, MAH 33881-1/3, and DS.9. The most infected heads had two lines DANKO 6 (2014), DL 446/08, and cultivar Meloman. The lowest FDK proportions (weight and number) were observed for two low FHB infected wheat lines. FHB-susceptible wheat lines showed only medium FDK values. Low FDK's were found only for two low FHB-infected triticale: DS.9 and BOHD 1025-2. The highest kernel damage was observed in line DANKO 9 (2013) that showed only weak FHB symptoms. FDK was also high in three lines, which showed the highest FHB index among triticale lines DANKO 6 (2014), DL 446/08, and BOHD 1062-2.

Reduction of grain yield per head was the highest in triticale line DANKO 6 (2014) as well as wheat line DL 325/11/3. These lines showed high levels of head infection and kernel damage. This was also noted in lines DANKO 9 (2013) and LD 121/08. The second line showed low head and kernel infection. Line DANKO 9 (2013) had the lowest average grain yield per head in control plots while line DANKO 6 (2014) the highest. Line LD 121/08 had medium yield. Low reduction of grain yield was found in resistant wheat lines and low-infected triticale BOH 534-4 and DS.9 as well as in medium-infected line BOH 537-2. Lines DS.9 and BOH 537-2 had average grain yield per head in control plots below mean value while line BOH 534-4 above mean value. Reduction of 1000 kernel weight was the highest in the susceptible wheat line DL 325/11/3 and the lowest in resistant wheat lines and resistant triticale line DS.9.

**Table 2.** Fusarium head blight index (%), *Fusarium*-damaged kernels (weight. number) (%) and reduction of yield, kernel number, and TKW (%) for 15 winter triticale and three winter wheat lines.

| Line | FHBi | FDKw | FDK# | Yield Reduction | Kernel Number Reduction | TKW Reduction |
|---|---|---|---|---|---|---|
| DL 325/11/3 [a] | 32.6 a | 25.7 abcd | 22.2 abcd | 41.9 ab | 27.1 | 22.7 a |
| Meloman | 14.0 bcd | 22.8 bcde | 18.5 bcde | 33.3 abcde | 25.5 | 10.8 bcde |
| DANKO 6 (2014) | 13.9 b | 31.0 ab | 25.2 ab | 44.4 a | 36.4 | 18.7 abc |
| DL 446/08 | 13.5 bc | 30.8 ab | 26.5 ab | 29.7 cde | 23.3 | 13.8 bcd |
| BOHD 1062-2 | 11.6 bcde | 28.0 abc | 24.8 abc | 27.0 cde | 20.8 | 14.7 bcd |
| BOH 898-1 | 10.7 bcdef | 23.1 abcde | 18.1 abcde | 24.3 cde | 17.4 | 10.5 bcde |
| BOH 537-2 | 10.2 bcdef | 20.8 abcde | 18.1 bcde | 23.5 ef | 16.1 | 11.4 bcde |
| DL 593/07 | 9.5 cdef | 23.6 abcde | 19.6 abcde | 29.0 bcde | 22.0 | 11.8 cdef |
| MAH 33544-4 | 9.4 bcdef | 18.0 cde | 13.9 cde | 28.0 abcde | 19.1 | 14.3 abc |
| DS.1238 | 9.4 bcdef | 24.7 abcde | 21.2 abcde | 26.9 cde | 19.8 | 12.3 bcd |
| DANKO 9 (2013) | 9.1 ef | 33.6 a | 28.6 a | 35.6 abc | 22.6 | 16.3 abc |
| LD 121/08 | 9.0 ef | 18.6 bcde | 15.6 cde | 35.8 abcd | 27.9 | 13.8 bcd |
| BOHD 1025-2 | 8.6 def | 14.4 e | 11.8 e | 26.0 cde | 17.5 | 13.2 bcd |
| BOH 534-4 | 8.4 ef | 21.3 abcde | 17.6 bcde | 23.9 ef | 15.8 | 12.0 bcd |
| MAH 33881-1/3 | 7.1 f | 22.7 abcde | 19.1 abcde | 25.6 cde | 15.1 | 16.7 ab |
| DS.9 | 7.1 f | 16.7 de | 13.0 de | 23.4 def | 21.0 | 8.0 ef |
| S 10 [a] | 4.0 g | 4.0 f | 3.4 f | 14.4 g | 13.1 | 4.5 f |
| S 32 [a] | 3.9 g | 8.7 f | 7.1 f | 16.0 fg | 13.2 | 7.6 def |
| Means | 10.7 | 18.1 | 21.6 | 28.3 | 20.8 | 12.9 |

[a] wheat; those marked with the same letter are not significantly different at *p* < 0.05 according to Fisher's least significant difference (LSD) test performed on transformed variables; means ranked by FHBi values.

Ergosterol concentration was the highest in grain of susceptible wheat line DL 325/11/3 (Table 3). It was twice as high as in two triticale lines that had the highest concentrations of this metabolite (DANKO 6 (2014) and BOHD 1062-2). These lines had also high kernel damage. Line DANKO 9 (2013) exhibiting highest kernel damage had medium concentration of ERG in grain. The lowest ERG content was found in grain of five triticale lines DS. 9, BOH 898-1, BOHD 1025-2, BOH 534-4, and BOH 537-2. FHB-resistant wheat lines had medium concentration of ERG in grain.

Deoxynivalenol accumulated the highest amount in grain of the triticale line DANKO 6 (2014) and in the wheat line DL 325/11/3. Four triticale lines, BOHD 1062-2, DL 446/08, DANKO 9 (2013) and DS.1238, also had high amounts of DON in grain. The lowest concentration of DON was detected in the grain of resistant wheat lines and two triticale lines BOH 534-4 and BOHD 1025-2. Nivalenol was present mainly in the grain of four triticale lines: DANKO 6 (2014), BOHD 1062-2, DL 446/08, DANKO 9 (2013), and MAH 33881-1/3. The lowest concentration of NIV was detected in the grain of resistant wheat lines and two triticale lines DS.9 and BOHD 1025-2. 3AcDON was detected mainly in susceptible wheat line and triticale line DANKO 6 (2014), that accumulated high amount of trichothecene toxins. Of the total amount of the four trichothecenes, the highest was in the grain of three triticale lines DANKO 6 (2014), BOHD 1062-2, and DL 446/08, and the lowest was in the grain of resistant wheat lines. The wheat line "DL 325/11/3" had a medium amount of type B trichothecenes in grain. Four triticale lines, BOH 898-1, DS. 9, BOH 534-", and BOHD 1025-2, accumulated the lowest amounts of type B trichothecenes.

Differences in ZEN concentrations among triticale lines had low significance. The lowest amount of ZEN was found in the grain of the DS.9 line. Considerably lower amounts of ZEN accumulated only in the grain of resistant wheat lines.

**Table 3.** Concentration of ergosterol (mg/kg), type B trichothecenes (DON, 3AcDON, 15AcDON, NIV, and TCT B) (mg/kg) and zearalenone (mg/kg) in grain of 15 winter triticale and three winter wheat lines.

| Line | ERG | DON | 3Ac DON | 15Ac DON | NIV | TCT B [b] | ZEN |
|---|---|---|---|---|---|---|---|
| DANKO 6 (2014) | 13.0 b | 14.327 a | 0.426 ab | 0.081 | 7.653 ab | 22.487 a | 0.966 ab |
| BOHD 1062-2 | 12.1 bc | 10.207 ab | 0.327 abcd | 0.012 | 10.966 a | 21.511 ab | 1.443 abc |
| DL 446/08 | 9.4 bcde | 10.882 abcd | 0.374 abc | 0.050 | 10.197 a | 21.503 ab | 0.956 ab |
| DANKO 9 (2013) | 8.6 bcde | 10.944 abc | 0.321 abcd | 0.043 | 7.838 a | 19.146 abc | 1.169 a |
| MAH 33881-1/3 | 6.4 b–g | 9.380 abcd | 0.251 abcde | 0.046 | 7.974 a | 17.652 abc | 0.791 abcd |
| DL 325/11/3 [a] | 28.3 a | 11.887 ab | 0.461 a | 0.021 | 4.930 abcd | 17.298 abc | 0.833 ab |
| DS.1238 | 7.5 b–g | 10.549 abcde | 0.335 abcde | 0.072 | 4.981 abcd | 15.937 abcd | 1.238 a |
| Meloman | 9.5 bcd | 9.089 abcde | 0.214 b–g | 0.064 | 5.733 abc | 15.101 abcd | 0.999 abcd |
| DL 593/07 | 6.2 b–g | 7.278 bcdef | 0.224 bcdef | 0.070 | 5.856 abcd | 13.429 bcde | 0.921 abc |
| MAH 33544-4 | 5.8 b–g | 5.204 cdef | 0.161 c–g | 0.015 | 6.301 abc | 11.681 cde | 0.635 abc |
| BOH 537-2 | 5.4 defg | 6.447 bcdef | 0.139 defg | 0.031 | 3.657 bcde | 10.273 def | 0.757 ab |
| LD 121/08 | 5.7 b–g | 4.338 f | 0.165 c–g | 0.026 | 4.611 bcde | 9.139 ef | 0.763 bcd |
| BOH 898-1 | 3.7 fg | 5.387 ef | 0.182 c–g | 0.026 | 2.958 def | 8.553 ef | 0.720 abc |
| DS.9 | 4.9 efg | 5.111 def | 0.137 defg | 0.019 | 2.888 efg | 8.156 ef | 0.529 bcd |
| BOH 534-4 | 4.9 cdefg | 3.977 f | 0.173 c–g | 0.019 | 3.440 cde | 7.609 ef | 0.846 abc |
| BOHD 1025-2 | 4.2 g | 3.086 fg | 0.116 efg | 0.008 | 2.492 efg | 5.701 f | 0.775 abcd |
| S 32 [a] | 7.5 b–g | 1.161 h | 0.045 g | 0 | 1.362 fg | 2.567 g | 0.061 d |
| S 10 [a] | 7.7 b–f | 1.398 gh | 0.053 fg | 0.021 | 0.976 g | 2.448 g | 0.092 cd |
| Means | 8.4 | 7.258 | 0.228 | 0.035 | 5.267 | 12.788 | 0.805 |

[a] wheat; [b] sum of DON, 3AcDON, 15AcDON, and NIV; means marked with the same letter are not significantly different at $p < 0.05$ according to Fisher's LSD test performed on transformed variables; means ranked by TCT B concentration.

The Fusarium head blight index correlated significantly with other variables except 15AcDON and NIV (Table 4). Coefficients had high values (>0.600) but were low for the sum of type B trichothecenes and ZEN concentrations. *Fusarium*-damaged kernel proportions (i.e., weight and number) correlated highly significantly with reductions in yield components and concentration of mycotoxins. The highest values had coefficients of correlations with TCT B and ZEN. The FDKs did not correlate with the ERG amount in grain. This resulted from higher than expected FDK value concentrations of ERG in grain of the wheat lines. For sole triticale lines, FDKs correlated significantly with ERG (0.768 and 0.759). Reductions in the yield components correlated significantly with concentrations of ERG and mycotoxins. The highest were coefficients of correlation with DON and 3AcDON, the lowest with ERG and NIV.

Ergosterol concentration in grain correlated significantly with DON and 3AcDON and did not correlate with the amounts NIV and ZEN. Trichothecene toxins correlated significantly with each other and with ZEN. The highest values had coefficients of correlation DON versus 3AcDON and DON versus NIV.

Analysis of variance of the FHB index showed a very high effect of the triticale/wheat line and no effect of year (random) and location (Table 5). No interaction year × line was observed. Highly significant interactions for year × location and location × line were found. Similarly, for FDKw and FDK#, the effect of line was highly significant as well year × location interaction. Interactions for location × line were not significant for FDK. For reductions in yield component, the effects of line were significant for grain yield and TKW but not for kernel number. Year × location interactions were significant for all components (Table 6).

**Table 4.** Coefficients of correlation between Fusarium head blight index, *Fusarium*-damaged kernels proportion (weight, number), reductions in yield components and concentration of ergosterol and mycotoxins in grain of 15 winter triticale and three winter wheat lines.

| Variables | FHBi | FDKw | FDK # | Yield Red. | Kernel# Red. | TKW Red. | ERG | DON | 3Ac DON | 15Ac DON | NIV | TCT B |
|---|---|---|---|---|---|---|---|---|---|---|---|---|
| FDKw | 0.611 ** | | | | | | | | | | | |
| FDK# | 0.606 ** | 0.996 | | | | | | | | | | |
| Yield red. | 0.763 | 0.753 | 0.766 | | | | | | | | | |
| Kernel# red. | 0.645 ** | 0.609 ** | 0.628 ** | 0.915 | | | | | | | | |
| TKW red. | 0.740 | 0.774 | 0.770 | 0.844 | 0.612 ** | | | | | | | |
| ERG | 0.698 | 0.362 $^{ns}$ | 0.326 $^{ns}$ | 0.532 * | 0.526 * | 0.526 * | | | | | | |
| DON | 0.673 ** | 0.865 | 0.855 | 0.773 | 0.699 | 0.750 | 0.656 ** | | | | | |
| 3AcDON | 0.769 | 0.837 | 0.822 | 0.787 | 0.700 | 0.806 | 0.730 | 0.949 | | | | |
| 15AcDON | 0.227 $^{ns}$ | 0.531 * | 0.540 * | 0.501 * | 0.543 * | 0.269 $^{ns}$ | 0.216 $^{ns}$ | 0.674 ** | 0.540 * | | | |
| NIV | 0.376 $^{ns}$ | 0.789 | 0.773 | 0.549 * | 0.479 * | 0.632 ** | 0.432 $^{ns}$ | 0.773 | 0.724 | 0.413 $^{ns}$ | | |
| TCT B | 0.584 * | 0.884 | 0.870 | 0.721 | 0.644 ** | 0.744 | 0.598 ** | 0.960 | 0.908 | 0.599 ** | 0.920 | |
| ZEN | 0.484 * | 0.879 | 0.864 | 0.627 ** | 0.506 * | 0.635 ** | 0.276 $^{ns}$ | 0.753 | 0.717 | 0.483 * | 0.750 | 0.797 |

Coefficients significant at $p < 0.001$, except when marked with *, **, $^{ns}$—significant at $p < 0.05$, 0.01 or non-significant, respectively.

**Table 5.** Analysis of variance of Fusarium head blight index (FBI) and *Fusarium*-damaged kernels percentage (weight, number).

| Source | DF | FHBi | | FDKw | | FDK# | |
|---|---|---|---|---|---|---|---|
| | | Mean Squares | F | Mean Squares | F | Mean Squares | F |
| Year | 2 | 91.803 | 6.178 | 73.178 | 1.001 | 54.475 | 1.155 |
| Location | 1 | 1.830 | 0.122 | 170.582 | 2.306 | 298.949 | 6.259 * |
| Line | 17 | 7.071 | 16.109 *** | 12.880 | 6.355 *** | 15.348 | 7.110 *** |
| Year × Location | 2 | 15.026 | 24.799 *** | 73.981 | 25.486 *** | 47.765 | 17.377 *** |
| Year × Line | 34 | 0.439 | 0.724 | 2.027 | 0.698 | 2.159 | 0.785 |
| Location × Line | 17 | 4.087 | 6.744 *** | 3.482 | 1.200 | 4.565 | 1.661 |
| Year × Location × Line | 34 | 0.606 | 1.449 | 2.903 | 6.762 *** | 2.749 | 5.223 *** |
| Error | 162 | 0.418 | | 0.429 | | 0.526 | |

***, * significant at $p < 0.001$ and 0.05, respectively.

**Table 6.** Analysis of variance of reductions in yield per head, kernel number in head, and 1000 kernel weight.

| Source | DF | Yield | | Kernel Number | | TKW | |
|---|---|---|---|---|---|---|---|
| | | Mean Squares | F | Mean Squares | F | Mean Squares | F |
| Year | 2 | 7.919 | 0.038 | 25.939 | 0.175 | 9.829 | 0.179 |
| Location | 1 | 263.319 | 1.285 | 94.387 | 0.641 | 317.916 | 5.845 * |
| Line | 17 | 16.187 | 3.579 *** | 11.723 | 1.856 | 8.286 | 2.545 ** |
| Year × Location | 2 | 204.966 | 56.587 *** | 147.266 | 28.703 *** | 54.392 | 20.617 *** |
| Year × Line | 34 | 4.522 | 1.249 | 6.316 | 1.231 | 3.256 | 1.234 |
| Location × Line | 17 | 6.666 | 1.840 | 10.997 | 2.143 * | 3.795 | 1.438 |
| Year × Location × Line | 34 | 3.622 | 2.924 *** | 5.131 | 3.929 *** | 2.638 | 5.634 *** |
| Error | 162 | 1.239 | | 1.306 | | 0.468 | |

***, **, * significant at $p < 0.001$, 0.01 and 0.05, respectively.

We found highly significant effects of year on concentration of all analyzed toxins (Tables 7 and 8). Location had a highly significant effect on concentration of trichothecenes and ZEN but only weak one on ERG concentration.

**Table 7.** Analysis of variance of concentration of ergosterol, DON, 3ACDON, and 15AcDON in grain.

| Source | DF | ERG | | DON | | 3AcDON | | 15AcDON | |
|---|---|---|---|---|---|---|---|---|---|
| | | **MS** | **F** | **MS** | **F** | **MS** | **F** | **MS** | **F** |
| Year | 2 | 1.154 | 25.786 *** | 3.634 | 87.057 *** | 0.200 | 70.513 *** | 0.008 | 10.253 *** |
| Location | 1 | 0.193 | 4.311 * | 2.087 | 49.989 *** | 0.031 | 11.008 *** | 0.018 | 24.035 *** |
| Line | 17 | 0.187 | 4.190 *** | 0.259 | 6.203 *** | 0.008 | 2.876 *** | 0.000 | 0.635 |
| Error | 87 | 0.045 | | 0.042 | | 0.003 | | 0.001 | |

***, * significant at $p < 0.001$ and 0.05, respectively

Location mean squares for 15AcDON, NIV, and ZEN were higher than year mean squares. These toxins were mainly detected in samples from Radzików (15AcDON) or Cerekwica (NIV, ZEN) (Figure 2). The effect of the triticale line was highly significant for ERG and trichothecenes (except 15AcDON) and low significant for ZEN.

**Table 8.** Analysis of variance of concentration of nivalenol, trichothecenes B (sum of DON, 3AcDON, 15AcDON, and NIV) and zearalenone in grain.

| Source | DF | NIV | | TCT B | | ZEN | |
|---|---|---|---|---|---|---|---|
| | | **MS** | **F** | **MS** | **F** | **MS** | **F** |
| Year | 2 | 0.825 | 31.975 *** | 2.526 | 74.300 *** | 13.499 | 18.648 *** |
| Location | 1 | 13.156 | 509.917 *** | 0.719 | 21.139 *** | 26.304 | 36.337 *** |
| Line | 17 | 0.147 | 5.707 *** | 0.352 | 10.356 *** | 1.293 | 1.787 * |
| Error | 87 | 0.026 | | 0.034 | | 0.724 | |

***, * significant at $p < 0.001$ and 0.05, respectively.

Multivariate PCA analysis showed that the highest FHB resistance explained by eight variables (FHBi, FDKw, yield, TKW, ERG, DON, NIV, and ZEN) was found in resistant wheat lines carrying the *Fhb1* gene and in two triticale lines (DS.9 and BOHD 1025-5) (Figure 3). Five other lines had also considerable FHB resistance (BOH 898-1, BOH 534-4, BOH 537-2, MAH 33544-1, and LD 121/08). More susceptible triticale lines showed high kernel damage and toxins accumulation. Susceptible wheat line DL 325/11/3 could be characterised by high Fusarium head blight severity (FHBi) and high ERG concentration.

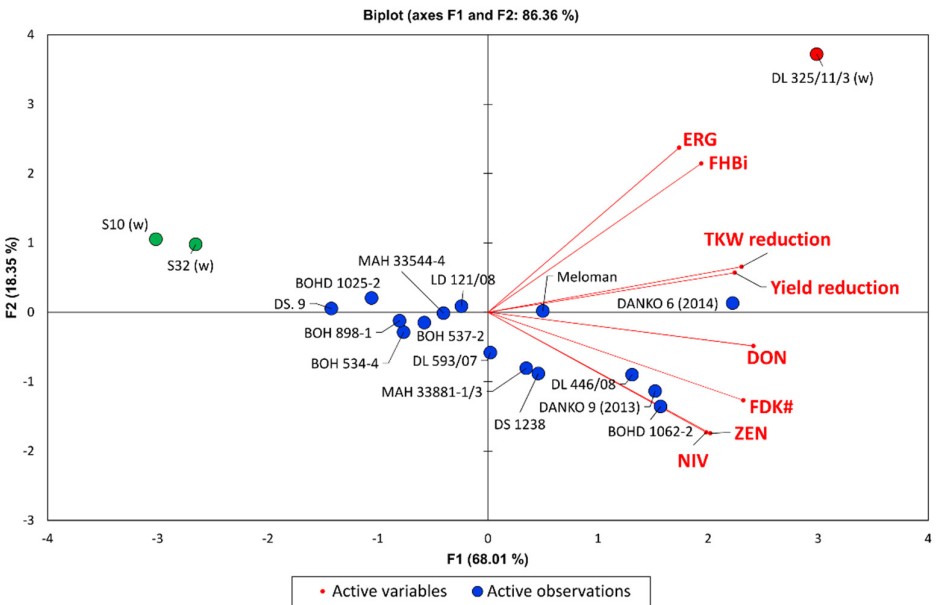

**Figure 3.** Biplot of principal component analysis (PCA) analysis of FHBi, FDK#, yield reduction, TKW reduction, ERG, DON, NIV, and ZEN for 15 winter triticale and 3 winter wheat lines (w).

The number of infection points (measure of type 1 resistance) was on average 1.89 (1.96 for triticale) at a range 1.00 (BOH 898-1, DANKO 9 (2013))–3.40 (BOH 537-2) (Table 9). Lines did not differ statistically significantly for type 1 resistance. However, the lowest type 1 resistance was found for wheat lines and three triticale lines (DS. 9, BOHD 1025-2 and DS 1238). The number of infected spikelets (measure of type 2 resistance) was on average 1.83 (1.84 for triticale) at a range 0.60 (wheat S32)–4.40 (DL 325/11 wheat). For triticale type 2 resistance range was 1.00 (BOH 537-2)–4.00 (DANKO 9 (2013)). Lines differed significantly for type 2 resistance; however, differences between most of the triticale lines were not significant.

**Table 9.** Type 1 and type 2 resistance to Fusarium head blight of 15 winter triticale and three winter wheat lines.

| Line | Type 1 [b] | Type 2 [b] | Type 1 + 2 [c] |
|---|---|---|---|
| DL 325/11/3 [a] | 1.78 | 3.47 a | 2.62 a |
| DANKO 6 (2014) | 2.31 | 2.18 ab | 2.24 ab |
| MAH 33881-1/3 | 2.13 | 2.23 ab | 2.18 abc |
| DANKO 9 (2013) | 1.95 | 2.33 b | 2.14 abc |
| BOH 534-4 | 2.20 | 2.03 b | 2.11 abc |
| Meloman | 2.16 | 2.02 b | 2.09 abc |
| BOHD 898-1 | 1.95 | 2.16 b | 2.05 abc |
| BOH 537-2 | 2.28 | 1.68 b | 1.98 bcde |
| LD 121/08 | 2.08 | 1.85 b | 1.96 abcde |
| DL 593/07 | 2.18 | 1.49 b | 1.83 bcde |
| BOH 1062-2 | 1.83 | 1.80 b | 1.81 bcde |
| DL 446/08 | 2.10 | 1.50 bc | 1.80 bcde |
| DS 1238 | 1.65 | 1.80 b | 1.73 cdef |
| MAH 33544-4 | 1.85 | 1.49 b | 1.67 bcde |
| BOHD 1025-2 | 1.45 | 1.49 bc | 1.47 defg |
| DS.9 | 1.26 | 1.57 b | 1.41 efg |
| S32 [a] | 1.45 | 0.98 cd | 1.22 fg |
| S10 [a] | 1.44 | 0.85 d | 1.14 g |
| Means | 1.89 | 1.83 | 1.86 |

[a] wheat; [b] average number of *Fusarium* infected spikelets; [c] average of type 1 and type 2; means marked with the same letter are not significantly different at $p < 0.05$ according to Fisher's LSD test performed on $\log_{10}$ transformed variables; means ranked by type 1 + 2.

Average resistance of type 1 and 2 was 1.86 (1.90 for triticale) at a range 1.03 (S10 wheat)–3.35 (BOH 537-2). The lines differed significantly for combined resistance of type 1 and 2. The most resistant were two wheat lines with *Fhb1* gene and two triticale lines DS.9 and BOHD 1025-2.

Type 1 and type 2 FHB resistance values were correlated with variables describing FHB resistance under field conditions (FHBi, FDK, ERG, and mycotoxin concentration) (Table 10). Type 1 resistance correlated weakly with type 2 resistance. It did not correlate with FHB index and ERG concentration, however correlated with other variables. The highest were coefficients for DON, FDK# and FDKw. Type 2 resistance correlated significantly with FHBi and other variables except ERG concentration. The highest were coefficients for FHBi, FDK3, and FDKw.

**Table 10.** Coefficients of correlation between type 1 and type resistances and Fusarium head blight index, *Fusarium*-damaged kernels proportion (weight, number) and concentration of ergosterol and mycotoxins in grain of 15 winter triticale and three winter wheat lines.

| Variables | Type 1 | Type 2 | Type 1 + 2 |
|---|---|---|---|
| Type 2 | 0.482 * | | |
| Type 1 + 2 | 0.759 | 0.931 | |
| FHBi | 0.347 ns | 0.748 | 0.730 |
| FDKw | 0.631 | 0.709 | 0.788 |
| FDK# | 0.633 | 0.725 | 0.795 |
| ERG | 0.276 ns | 0.151 ns | 0.249 ns |
| DON | 0.662 | 0.698 | 0.794 |
| NIV | 0.485 * | 0.654 | 0.702 |
| TCT B | 0.608 | 0.697 | 0.777 |
| ZEN | 0.540 ** | 0.679 | 0.722 |

Coefficients significant at $p < 0.001$, except when marked with *, **, ns—significant at $p < 0.05$, 0.01 or non-significant, respectively.

The highest correlation coefficients were found for the combined type 1 and type 2 resistances (Table 10). Similarly, ERG concentration did not correlated with type 1 + 2 resistance. The strongest were correlations with FDK# and FDKw as well as with DON and sum of type B trichothecenes.

Three triticale lines with the highest resistance of both types (1 and 2) showed also high resistance level of different types evaluated under field conditions (DS.9, BOHD 1025-2, MAH 33544-4) (Figure 3).

## 4. Discussion

Due to the increase in crop area and exposure to a variety of pathogens in triticale, there has been a breakdown in crop resistance against fungal diseases [48,49]. The most remarkable examples have been powdery mildew and yellow rust [50,51]. Recently, a decrease in triticale resistance to pathogens of the *Fusarium* genus have been observed. FHB outbreaks in wheat have become more serious and frequent in recent decades, possibly due to the changes in climate and agronomic practices [52]. Globally, FHB causes approximately 10–70% yield loss in epidemic years [53,54]. Because triticale is consumed mainly by farm animals, it is important to maintain good quality grains, especially in case of detrimental toxin content [55,56].

We identified triticale lines highly resistant to FHB and, in particular, to the accumulation of *Fusarium* toxins. However, some lines despite their low head infection accumulated considerable amounts of trichothecenes in grain, e.g., DANKO 6 (2014), BOHD 1062-2, DL 446/08, DANKO 9 (2013), and MAH 33881-1/3. All these lines (except MAH 33881-1/3) had high *Fusarium* kernel damage. We observed this problem previously when comparing wheat and triticale under the same conditions in other inoculation experiments [27]. According to conference presentations by Randhawa et al. (2013) and Langevin (2009) (cited by Randhawa et al. [3]), screening of a large number of triticale accessions resulted in only a few lines with a good level of FHB resistance. Some lines showed higher DON accumulation than expected from low head infection. Langevin et al. (2009) speculated that the higher DON content in triticale grain might be caused by a more fragile pericarp during the initial development of the triticale seed.

Research has shown that environmental conditions significantly affect the development of FHB and the accumulation of toxins in the grain [11,57]. Fusarium head blight severity, kernel damage, and concentration of *Fusarium* metabolites were significantly affected by the experimental year and the location. This study on resistance to FHB and accumulation of *Fusarium* toxins was conducted over three years in two locations. To maintain humidity during inoculation and after inoculation, mist irrigation was used in Cerekwica. The infection of heads was similar in Cerekwica and Radzików, but the other parameters examined—percentage of FDK, number and weight of grains per head, and the reduction in

the yield structure parameters—were much higher in Poznań than in Radzików. Similarly, the amount of toxins: type B trichothecenes and zearalenone was higher in Cerekwica.

Weather conditions in 2016 were similar in Cerekwica and Radzików (Supplementary Materials Table S1). Rainfall before anthesis (May) was low and higher (approximately 50%) during and after anthesis (June). This resulted in similar head infections in both locations. Next, in July, rainfall in Cerekwica was much higher than in Radzików which (accompanied by higher temperature) resulted in higher kernel damage in the first location. In addition, toxin accumulation in grain in Cerekwica was higher. In 2017, the weather in May was similar in both locations, and in June, rainfall in Radzików was twice as high as in Cerekwica. This led to a higher head infection in Radzików. Similar to July 2016, rainfall in Cerekwica was double that of Radzików. This caused very high kernel damage in Cerekwica. However, the amounts of trichothecenes in grain in both locations were similar and higher than in 2016. We observed differences in the accumulation of DON and NIV in both locations. DON was mainly found in grain from Radzików and NIV mainly in samples from Cerekwica. Weather in 2018 was unfavourable for FHB development, particularly in Cerekwica. Rainfall in June was low compared to previous years. In both locations, head infection was low, even despite application of mist irrigation in Cerekwica. Kernel damage was lower than in previous years and similar in locations. The same was found for trichothecenes and ZEN. As in 2017, we observed the opposite results for DON and NIV concentrations.

Probably observed differences in accumulation of DON and NIV in two locations were result of competition between isolates of different chemotypes. Competition between *F. culmorum* and *F. graminearum* species was described by Van der Ohe and Miedaner [58]. One isolate of *F. graminearum* was of the NIV chemotype and this isolate showed similar pathogenicity to isolates of the DON chemotype. Mixture of DON+NIV chemotypes had stable pathogenicity (head infection), but varied in mycotoxin production in experimental environments. The NIV chemotype is generally considered less aggressive than DON (3ADON, 15ADON) chemotypes [59,60]. Results of our experiments showed that it could produce considerable amounts of NIV even in mixture with a more aggressive isolates of the 3ADON chemotype. However, it occurred only in Cerekwica, where application of mist irrigation created conditions more favourable for FHB development.

Under the above variable conditions, some triticale lines showed stable reactions (i.e., were ranked consistently resistant in all environments) for most variables describing FHB resistance. The most stable were the resistant lines BOHD 1025-5, DS.9 and MAH 33544-4. Line BOH 534-4 had a stable reaction to head infection and accumulation of ERG and trichothecenes (low infection and low accumulation); however, less it was stable as regards kernel damage. Similarly, the BOH 898-1 line was less stable for head infection and kernel damage but stable for low ERG and trichothecenes accumulation.

Selection of FHB-resistant genotypes is more complicated in triticale than in wheat. Frequently *Fusarium* head infection is lower, but other FHB components are comparable to wheat, which is more susceptible. Low visible head infection can result in significant amounts of infected kernels and accumulation of toxins in grain [61,62]. Studies conducted on triticale lines showed that the assessment of resistance to FHB based only on head infection symptoms was only ineffective for the selection of resistant genotypes [61]. The results may differ in the locations of experiments as can be seen from the significance of the interaction between location × line. Reliable assessment of FHB resistance has to be associated with the evaluation of kernel damage and the amount of toxins accumulated in the grain. Ollier et al. [63] showed that in triticale, FHB severity symptoms on grain measured digitally as "whitened kernel surface" [64] had higher heritability coefficients than FHB symptoms on heads. This variable highly correlated with the mycotoxin content. We also observed much higher values of coefficients for correlations FDK versus mycotoxins than for FHB index versus mycotoxins.

We found that application of ERG content as a measure of *Fusarium* colonisation of grain is of limited use. Generally, ERG content weakly correlated with other FHB resistance

types—lack of correlation with kernel damage was observed. It resulted from the fact that ERG is the compound produced by different fungal species and is no specific for *Fusarium* genus [34]. Quantitative real time PCR seems to be more suitable for measuring *Fusarium* colonisation of grain as it is specific at genus and species level [65,66].

The introduction into triticale of genes associated with resistance to FHB is very desirable, although it is difficult due to the complex genome of this cereal. Only a few papers have been published on the genetics of FHB resistance in triticale [25,67,68]. In papers by Kalih et al. [25,68], 17 FHB-resistance QTLs were presented including six on the rye chromosomes. The most effective (34%) was the QTL on chromosome 4R. Dhariwal et al. [67] obtained similar results. Using single nucleotide polymorphism (SNP) genotyping, they identified 17 QTLs explaining more than 10% of the variability in FHB resistance. Seven of them were located on rye chromosomes (4R, 5R). The highest effect had four QTLs on chromosomes 1A, 2B, 4R, and 5R. Association mapping, carried out on varieties and breeding lines differentiated in origin, allowed the identification of QTLs on chromosomes 2A, 2B, 5B, and for the first time on 3R, with individual QTLs explaining variability in the range of 0.28–30.23% [69]. The authors highlight the possibility of increasing triticale resistance to FHB in the early generations using tools proposed by genomics.

Recently, an attempt was made to introduce FHB resistance from the spring wheat line CM-82036 into the triticale varieties [63,70]. The *Fhb1* gene was detected and validated in triticale background. Nine triticale lines with very high levels of FHB resistance were identified. QTL analysis of FHB resistance showed the presence of additional loci of resistance on chromosomes 2B, 5R, and 7A. QTL on chromosome 5R coincided with the location of the dwarfing gene *Ddw1*. Similar research on the introduction of the *Fhb1* gene into triticale was also carried out by the Institute of Plant Genetics group [71].

## 5. Conclusions

We found large variability of reaction of triticale lines to Fusarium head blight. Some of the lines, despite low head infection exhibited significant *Fusarium* kernel damage and high level of *Fusarium* toxin accumulation. However, lines combing all types of FHB resistance were identified. Two of them had the highest FHB resistance, however lower than wheat lines with the *Fhb1* gene used as a highly resistant control.

**Supplementary Materials:** The following are available online at https://www.mdpi.com/2073-4395/11/1/16/s1, Table S1: Air temperature and rainfall in May, June, and July of 2016, 2107, and 2018 in two experimental locations.

**Author Contributions:** Conceptualization, H.W. and T.G.; methodology, H.W., T.G., and P.O.; validation, T.G.; formal analysis, T.G.; investigation, H.W., T.G., P.O., D.W.-G., and A.T.; resources, H.W., T.G., and P.O.; data curation, H.W., T.G., and P.O.; writing—original draft preparation, H.W. and T.G.; writing—review and editing, H.W., T.G., P.O., D.W.-G., and A.T.; visualization, T.G.; project administration, H.W.; funding acquisition, H.W. All authors have read and agreed to the published version of the manuscript.

**Funding:** Research was supported by projects from the Ministry of Agriculture and Rural Development: "Evaluation of Fusarium head blight resistance types in winter triticale using phenotypic and metabolic markers", project no. 14, and "Identification, and application of phenotypic, metabolic and molecular markers in studies of types of resistance to Fusarium head blight in winter wheat accessions differing in resistance", project no. 6 (HOR hn 801-PB-13/16, HOR.hn.802.28.2017, HOR.hn.802.19.2018).

**Acknowledgments:** The Authors wish to thank triticale and wheat breeders from Hodowla Roślin Strzelce Ltd., Co.–IHAR-PIB Group and DANKO Hodowla Roślin Ltd. for providing triticale and wheat lines for the experiment.

**Conflicts of Interest:** The authors declare no conflict of interest.

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
