# Peer review of "Resistance to Fusarium Head Blight, Kernel Damage, and Concentration of Fusarium Mycotoxins in Grain of Winter Triticale (x Triticosecale Wittmack) Lines"

_agronomy, doi:10.3390/agronomy11010016_

Round 1

Reviewer 1 Report

In this manuscript, Góral et al. identified triticale lines highly resistant to FHB and, in particular, to the accumulation of mycotoxins based on resistance evaluation over two locations and three year data. Authors used different FHB component traits (including mycotoxins) and estimated ANOVA, correlation coefficient and PCA. This study is a good piece of research on FHB in triticale. Though, interest in the research is average but for triticale, its an important paper. Minor revisions based on comments on the pdf file are required.

Author Response

Replay to Review 1

Comments and Suggestions for Authors

In this manuscript, Góral et al. identified triticale lines highly resistant to FHB and, in particular, to the accumulation of mycotoxins based on resistance evaluation over two locations and three year data. Authors used different FHB component traits (including mycotoxins) and estimated ANOVA, correlation coefficient and PCA. This study is a good piece of research on FHB in triticale. Though, interest in the research is average but for triticale, its an important paper. Minor revisions based on comments on the pdf file are required.

Thank you for your review. We tried our best to revise the article in accordance with your comments.

In details:

Introduction was revised according to your remarks. Missing references were added.

Plant material: We provided pedigree of all lines and the list was converted to table, which is easier to read. We don’t have any information about resistance genes (to other diseases) in these lines. I have only some data on FHB resistance of some triticale cultivars present in pedigree form my experiments.

Inoculation procedure: Reference was added. More details were added to the description of threshing procedure.

Figures 1 and 2 were revised (also according to remarks of other rewiever).

Composition of Results was changed. ANOVA tables and description were moved below tables and description of lines.

Discussion was revised.

Best regards

Tomasz Góral

Reviewer 2 Report

Introduction and Results, need to be shrunk and made more efficient, and plain to read. Specifically, information about limits in diverse stocks fall in too many details, lines 78-100. Description of figures 1 and 2, lack of statistical description in the caption. Figure 1 is totally uninformative considering the standard deviation  

In Methods, not for each experiment, the number of replicates and sampling methodology are clearly identified 

In accord, Discussion requires to be modified.

Author Response

Replay to Review 2

Comments and Suggestions for Authors

Introduction and Results, need to be shrunk and made more efficient, and plain to read. Specifically, information about limits in diverse stocks fall in too many details, lines 78-100. Description of figures 1 and 2, lack of statistical description in the caption. Figure 1 is totally uninformative considering the standard deviation.   

In Methods, not for each experiment, the number of replicates and sampling methodology are clearly identified.  

In accord, Discussion requires to be modified.

Thank you for your review. We tried our best to revise the article in accordance with your comments.

In details:

Introduction and Results were rewritten. Information about limits was shortened.

Figures 1 and 2 were revised (also according to remarks of other rewievers).

Missing information in Methods was added.

Best regards

Tomasz Góral

Reviewer 3 Report

Reviewer Summary:

The authors provide a comprehensive evaluation of FHB susceptibility in 15 triticale breeding lines. The findings are novel and should be of interest to pathologists and breeders who read Agronomy. The experimental replication, three years and two locations, is especially commendable, and the writing is generally acceptable (see some minor corrections below, but a careful reading is also recommended). The work is scientifically sound and provides substantially new information. Some recommendations are made to clarify the interpretation of data and to provide a stronger discussion of the results reported.

Broad Comments:

  • The opening paragraphs introduce the multiple FHB resistance concepts (ie Type I-V), and the extensive data collection permits a detailed analysis of how breeding lines vary in their potential to have different resistance traits. However, this is not fully explored in the data analysis or the discussion. For example, FHB index is a good composite measure of susceptibility, but why not decompose this into incidence (% of spikes infected) and severity (% of symptomatic tissue on infected spikes) to contrast both type I and II resistance-related symptoms? There are some points made in the discussion regarding which phenotypic traits should be considered during selection, but the discussion would benefit significantly from explicitly revisiting these resistance concepts and exploring whether some lines exhibit one or more type of resistance.

  • The disease observations made in non-inoculated control plots should be included in the manuscript so the following two points can be evaluated. (1) How does yield reduction (%) compare to absolute yield – for example, do lines with low yield reduction normally yield more or less grain than those exhibiting high yield reduction; (2) Are the differences in toxin accumulation between sites due to changes in the relative fitness of the three inoculated isolates or more likely due to infection by background inoculum sources? More appropriate consideration of the control plots should allow for a more detailed analysis of these data and ultimately a better evaluation of the plant lines.

  • More discussion of why ergosterol was measured is needed. Why is this important to measure? Was this useful or not to evaluating breeding line performance?

Specific Comments:

Methods:

  • Please include some information about the cropping history of the field sites if possible. At the least, discuss whether the triticale lines were planted into the same field in three consecutive years or whether the experimental plots moved with a rotation.
  • What were the planting dates? Were fertilizers or herbicides applied? Was the field tilled? How similar was anthesis timing for all the lines?

Results:

  • Figure 1 and 2 are difficult to interpret. Removing the lines connecting observations at each location is recommended, and the same can be done with the mean values printed by each bar. Erg and Zen should be represented with distinct colors. A better approach may be to re-organize both plots, placing the variables of interest on the x-axis and with bars from the two locations paired side-by-side (with different shading) for each variable. This would aid readers in comparing the sites.

Discussion:

  • L387. Was only one site irrigated? Was this mentioned in the methods and doesn’t this have implications for the comparison of the two sites RE: similar infection but different toxin levels?
  • L408-411. What is meant by stable reaction? Low variance or standard deviation? Or consistently ranked most resistant?
  • L344. 3AcDOn -> 3AcDON; read closely for other errors
  • L364. breakdown in their resistance -> breakdown in crop resistance
  • L370. toxins -> toxin

Author Response

Replay to Review 3:

Comments and Suggestions for Authors

 Reviewer Summary:

The authors provide a comprehensive evaluation of FHB susceptibility in 15 triticale breeding lines. The findings are novel and should be of interest to pathologists and breeders who read Agronomy. The experimental replication, three years and two locations, is especially commendable, and the writing is generally acceptable (see some minor corrections below, but a careful reading is also recommended). The work is scientifically sound and provides substantially new information. Some recommendations are made to clarify the interpretation of data and to provide a stronger discussion of the results reported.

Thank you for your review. We tried our best to revise the article in accordance with your comments.

In details:

  • The opening paragraphs introduce the multiple FHB resistance concepts (ie Type I-V), and the extensive data collection permits a detailed analysis of how breeding lines vary in their potential to have different resistance traits. However, this is not fully explored in the data analysis or the discussion. For example, FHB index is a good composite measure of susceptibility, but why not decompose this into incidence (% of spikes infected) and severity (% of symptomatic tissue on infected spikes) to contrast both type I and II resistance-related symptoms? There are some points made in the discussion regarding which phenotypic traits should be considered during selection, but the discussion would benefit significantly from explicitly revisiting these resistance concepts and exploring whether some lines exhibit one or more type of resistance. 

In years 2015 - 2019 we made series of experiments on type 1 and type 2 FHB resistance in wheat and triticale. For type 2 we used point inoculation and scoring 21 days post inoculation. For type 1 we used spraying with spore suspension and scoring about 7 days post inoculation. Plots were placed in foliar tents with mist irrigation so it was possible to keep high humidity after inoculations.

Fortunately, in four years all triticale and wheat lines studied in this research were evaluated for type 1 and type. Therefore, we decided to add these results to the paper instead of dissecting field observations of FHB index.

  • The disease observations made in non-inoculated control plots should be included in the manuscript so the following two points can be evaluated. (1) How does yield reduction (%) compare to absolute yield – for example, do lines with low yield reduction normally yield more or less grain than those exhibiting high yield reduction; (2) Are the differences in toxin accumulation between sites due to changes in the relative fitness of the three inoculated isolates or more likely due to infection by background inoculum sources? More appropriate consideration of the control plots should allow for a more detailed analysis of these data and ultimately a better evaluation of the plant lines. 

We did not observed natural FHB infection on triticale heads in control plots, even in year 2017, which had weather the most favourable for FHB (added to the Results). In Cerekwica, control plots were not mist irrigated.

We did not evaluated yield per plot for our lines. Only yield per head and 1000 kernel weight. Therefore, it would be difficult to compare general yield of lines using yield per head. We do not know what was number of head per plot. However, I have included some information about yield per head in control plots.

As regard point (2) we included some comments on differences in toxin accumulation (DON versus NIV) in experimental sites. I presume it was the effect on different conditions caused by use of mist irrigation which helped development of less pathogenic NIV chomotype.

  • More discussion of why ergosterol was measured is needed. Why is this important to measure? Was this useful or not to evaluating breeding line performance? 

We added short information about ergosterol in Introduction and more discussion about its usefulness for measuring kernel colonisation by Fusarium fungi.

Methods:

  • Please include some information about the cropping history of the field sites if possible. At the least, discuss whether the triticale lines were planted into the same field in three consecutive years or whether the experimental plots moved with a rotation.
  • What were the planting dates? Were fertilizers or herbicides applied? Was the field tilled? How similar was anthesis timing for all the lines?

We added more details (as you suggested) to the description of field experiments.

Results:

  • Figure 1 and 2 are difficult to interpret. Removing the lines connecting observations at each location is recommended, and the same can be done with the mean values printed by each bar. Erg and Zen should be represented with distinct colors. A better approach may be to re-organize both plots, placing the variables of interest on the x-axis and with bars from the two locations paired side-by-side (with different shading) for each variable. This would aid readers in comparing the sites.

Figures were reorganized and locations were paired for each variable. We left different colors for locations because bars for variables are paired together and there is a space between variables. 

Discussion:

  • L387. Was only one site irrigated? Was this mentioned in the methods and doesn’t this have implications for the comparison of the two sites RE: similar infection but different toxin levels?

We added more details to the description of field experiment. Effect of mist irrigation was mentioned in discussion.

  • L408-411. What is meant by stable reaction? Low variance or standard deviation? Or consistently ranked most resistant?

By “stable reaction’ we meant  that particular line had for example low infected head or accumulated low amount of DON etc. in all environments  (3 years x 2 locations) comparing with other lines.

  • L344. 3AcDOn -> 3AcDON; read closely for other errors
  • L364. breakdown in their resistance -> breakdown in crop resistance
  • L370. toxins -> toxin

Corrected.

Best regards

Tomasz Góral
